# Ventilatory efficiency is superior to peak oxygen uptake for prediction of lung resection cardiovascular complications

Andrej Mazur[1,2,3], Kristian Brat[2,3,4], Pavel Homolka[2,3,5], Zdenek Merta[2,4], Michal Svoboda[2,6], Monika Bratova[2,4], Vladimir Sramek[1,2,3], Lyle J. Olson[7], Ivan Cundrle[1,2,3] *

**1** Department of Anesthesiology and Intensive Care, St. Anne's University Hospital Brno, Brno, Czech Republic, **2** Faculty of Medicine, Masaryk University, Brno, Czech Republic, **3** International Clinical Research Center, St. Anne's University Hospital Brno, Brno, Czech Republic, **4** Department of Respiratory Diseases, University Hospital Brno, Czech Republic, **5** Department of Sports Medicine and Rehabilitation, St. Anne's University Hospital, Brno, **6** Institute of Biostatistics and Analyses, Ltd., Brno, Czech Republic, **7** Department of Cardiovascular Diseases, Mayo Clinic, Rochester, MN, United States of America

* Ivan.Cundrle@seznam.cz

**Data Availability Statement:** All relevant data are within the paper and its Supporting Information files.

## Abstract

### Introduction

Ventilatory efficiency ($V_E/VCO_2$ slope) has been shown superior to peak oxygen consumption ($VO_2$) for prediction of post-operative pulmonary complications in patients undergoing thoracotomy. $V_E/VCO_2$ slope is determined by ventilatory drive and ventilation/perfusion mismatch whereas $VO_2$ is related to cardiac output and arteriovenous oxygen difference. We hypothesized pre-operative $VO_2$ predicts post-operative cardiovascular complications in patients undergoing lung resection.

### Methods

Lung resection candidates from a published study were evaluated by post-hoc analysis. All of the patients underwent preoperative cardiopulmonary exercise testing. Post-operative cardiovascular complications were assessed during the first 30 post-operative days or hospital stay. One-way analysis of variance or the Kruskal–Wallis test, and multivariate logistic regression were used for statistical analysis and data summarized as median (IQR).

### Results

Of 353 subjects, 30 (9%) developed pulmonary complications only (excluded from further analysis), while 78 subjects (22%) developed cardiovascular complications and were divided into two groups for analysis: cardiovascular only (n = 49) and cardiovascular with pulmonary complications (n = 29). Compared to patients without complications (n = 245), peak $VO_2$ was significantly lower in the cardiovascular with pulmonary complications group [19.9 ml/kg/min (16.5–25) vs. 16.3 ml/kg/min (15–20.3); $P<0.01$] but not in the cardiovascular only complications group [19.9 ml/kg/min (16.5–25) vs 19.0 ml/kg/min (16–23.1); $P = 0.18$]. In contrast, $V_E/VCO_2$ slope was significantly higher in both cardiovascular only [29

**Funding:** - IC and KB received the award - Supported by Ministry of Health of the Czech Republic research grant [NU21-06-00086] - Ministry of Health of the Czech Republic - https://www.isvavai.cz/cep?s=jednoduche-vyhledavani&ss=detail&h=NU21-06-00086 - The funders had no role in study design, data collection and analysis, decision to publish, or preparation of the manuscript.

**Competing interests:** The authors have declared that no competing interests exist.

(25–33) vs. 31 (27–37); $P = 0.05$] and cardiovascular with pulmonary complication groups [29 (25–33) vs. 37 (34–42); $P<0.01$)]. Logistic regression analysis showed $V_E/VCO_2$ slope [OR = 1.06; 95%CI (1.01–1.11); $P = 0.01$; AUC = 0.74], but not peak $VO_2$ to be independently associated with post-operative cardiovascular complications.

## Conclusion

$V_E/VCO_2$ slope is superior to peak $VO_2$ for prediction of post-operative cardiovascular complications in lung resection candidates.

## Introduction

Peak oxygen consumption ($VO_2$) is recommended by both American and European guidelines for pre-operative risk stratification of lung resection candidates [1,2]. However, over the last decade, several studies have shown peak $VO_2$ to be a poor predictor of post-operative pulmonary complications [3] and inferior to ventilatory efficiency ($V_E/VCO_2$ slope) [4,5].

By the alveolar gas equation ventilatory efficiency is defined as $V_E/VCO_2 = 863 / (PaCO_2$ x $(1—V_D/V_T))$ [6]. Therefore, $V_E/VCO_2$ is increased by lower $PaCO_2$ (hyperventilation) or increase of the $V_D/V_T$ ratio (ventilation/perfusion mismatch or ineffective breathing pattern with rapid and shallow breathing) which may explain its superior prediction of post-operative pulmonary complications [6].

In contrast, determinants of $VO_2$ include cardiac output [7] and arteriovenous oxygen difference. Peak $VO_2$ is considered a cardiovascular fitness parameter [8] and was shown to be a major risk factor for post-operative cardiovascular complications [9]. Hence, an association with cardiovascular complications might be expected.

We hypothesized peak exercise $VO_2$ predicts post-operative cardiovascular complications in patients undergoing thoracic surgery. Accordingly, the aim of this study was to compare pre-operative peak exercise $VO_2$ and $V_E/VCO_2$ slope in the prediction of post-operative cardiovascular complications in patients after lung resection surgery.

## Methods

### Subject selection

Subjects included were consecutive patients who underwent lung resection surgery. Inclusion criteria were age ≥18 years and ability to undergo cardiopulmonary exercise testing (CPET). Exclusion criteria were contraindication for lung resection because of inoperability or low peak predicted post-operative $VO_2$ [$VO_2<10$ml/kg/min or <35% predicted together with predicted post-operative forced expiratory volume at one second ($FEV_1$) <30% and diffusing capacity for carbon monoxide ($DL_{CO}$) <30% $^2$)] and development of pulmonary complications only.

### Study design

This is a post-hoc analysis of a previously published prospective multicenter study which evaluated pre-operative rest ventilatory parameters as predictors of post-operative pulmonary complications in lung resection candidates [10]. All of the included patients underwent preoperative CPET and pulmonary function tests. Post-operative cardiovascular complications were prospectively accessed from the first 30 post-operative days or from the hospital stay. The

study was conducted in accordance with the declaration of Helsinki. All participants provided written informed consent and the study was approved by the local Ethics Committee of St. Anne's University Hospital in Brno (reference No. 19JS/2017, date of approval April 12, 2017; reference No. 2G/2018, date of approval March 21st, 2018) and by the local Ethics Committee of the University Hospital Brno (reference No. 150617/EK, date of approval June 19th, 2017). The study was registered at ClinicalTrials.gov (NCT03498352) and the manuscript adheres to the applicable STROBE guidelines.

## Cardiopulmonary exercise testing

Exercise capacity was assessed by CPET as described previously [10]. Subjects underwent preoperative CPET (symptom-limited) using a bicycle ergometer (Ergoline, Ergometrics 800®, Bitz, Germany). Expired gas analysis was done with the use of a PowerCube-Ergo® cardiopulmonary exercise system (Ganshorn Medizin Electronic GmbH, Niederlauer, Germany). Arterial blood gas analysis was done twice (at rest and peak exercise) using the ABL90 Flex Plus® (Radiometer Medical ApS, Denmark) device. The CPET protocol included a warm-up phase without resistance (60 seconds) followed by a linearly increasing (15 W/min) ramp protocol. Measured parameters included $VO_2$, carbon dioxide output ($VCO_2$), breathing frequency ($f_b$), tidal volume ($V_T$), minute ventilation ($V_E$) and $P_{ET}CO_2$. The derived CPET parameters included dead space to tidal volume ratio ($V_D/V_T$), respiratory exchange ratio (RER) and $V_E/VCO_2$ slope.

## Pulmonary function tests

Spirometry was performed using the ZAN100 device (nSpire Health, Inc., USA) and $DL_{CO}$ using the PowerCube Diffusion+ device (Ganshorn Medizin Electronic GmbH, Germany). Analyzed parameters include forced vital capacity (FVC), forced expiratory volume in one second ($FEV_1$) and $FEV_1/FVC$ ratio and were reported as percent (%) of predicted value.

## Cardiovascular complications

Cardiovascular complications were defined as previously recommended [11] and prospectively assessed from the first 30 post-operative days or from the hospital stay including arrhythmias (atrial fibrillation, supraventricular tachycardia–duration $\geq$ 10 seconds [12]); hypotension (catecholamine administration or parenteral fluids >200ml/h); heart failure (recently elevated serum brain natriuretic peptide and/or administration of inotropes); pulmonary edema (chest X-ray + clinical signs); pulmonary embolism (CT-angiography or highly suspicious right heart echocardiography signs); myocardial infarction/minimal myocardial lesion (coronary angiography or new ECG signs with serum positive troponin), stroke (newly developed neurological deficit with CT signs of brain infarction of ischemic or hemorrhagic etiology [13]) and cardiopulmonary resuscitation.

## Pulmonary complications

Pulmonary complications were assessed as previously described by our group [10] and others [4,14] and included pneumonia; atelectasis; respiratory failure requiring mechanical ventilation; adult respiratory distress syndrome and tracheostomy. ICU length of stay (LOS), hospital LOS and 30-day mortality were also monitored. Surgical Mortality Probability Model (S-MPM), an ASA (American Society of Anesthesiologists) physical status adjusted for surgery risk class and emergency status [15], was used to allow patient's co-morbidities comparison.

## Statistics

The Shapiro-Wilk test was used to evaluate normality. One-way analysis of variance or the Kruskal–Wallis test by ranks followed by either post-hoc Tukey HSD test or Mann–Whitney U test (as appropriate) were used to test for differences among the groups. Chi-square test was used to compare categorical variables. Univariate logistic regression was done for parameters significantly different between groups. The Spearman rank test was performed to evaluate parameters with significant correlation. Parameters with correlation coefficients $<\pm 0.5$ were used for multivariate stepwise logistic and linear regression. A multivariate stepwise logistic regression model was constructed to analyze which parameters were independently associated with the development of cardiovascular complications, ICU readmission and 30-day mortality. To evaluate model performance receiver operating characteristic (ROC) analyses were performed. Multivariate stepwise linear regression analyses (using same parameters as for multivariate logistic regression) were performed to evaluate the association of peak $VO_2$ and $V_E/VCO_2$ slope with hospital and ICU LOS. Data are summarized as mean $\pm$ SD or median (IQR) as appropriate; $P$ values $<0.05$ were considered statistically significant. Statistica software 12.0 (StatSoft Inc., Prague, Czech Republic) and IBM SPSS Statistics 25.0 (IBM Corp, USA) were used for analysis.

## Results

Data from 353 lung resection surgery candidates from our previous study [10] were analyzed. Thirty patients (9%) developed pulmonary complications only and were excluded; 78 patients (22%) developed cardiovascular complications and 245 patients (69%) had no complications. Comparison of patients with and without cardiovascular complications is shown in S1 Table. Out of the 78 patients with cardiovascular complications, 49 patients developed cardiovascular complications only and 29 patients developed cardiovascular with pulmonary complications. Cardiovascular and pulmonary complications often coexist [16], and despite prospective assessment of complications, we were not able to fully distinguish which complications were primary and which were secondary. To mitigate this, we divided our cardiovascular complications subjects into 2 groups–cardiovascular only and cardiovascular with pulmonary complications (frequency of cardiovascular complications was nearly the same in both groups Table 1) and analyzed them separately.

Subject and surgery characteristics are shown in Table 2. Subjects in both the post-operative cardiovascular complications groups were significantly older and had higher S-MPM and comorbidities. Hospital and ICU LOS was significantly longer and frequency of ICU readmissions higher in both cardiovascular complication groups. The 30-day mortality was significantly higher in the cardiovascular with pulmonary complications group.

**Table 1. Cardiovascular complications: Type and frequency (n = 78).**

|  | Cardiovascular only (n = 49) | With pulmonary (n = 29) | P |
|---|---|---|---|
| Lung edema n (%) | 0 | 2 (7) | 0.14 |
| Pulmonary embolism n (%) | 3 (6) | 1 (3) | 1.00 |
| Arrhythmia n (%) | 26 (53) | 21 (72) | 0.10 |
| Hypotension n (%) | 26 (53) | 19 (66) | 0.35 |
| Heart failure n (%) | 0 | 2 (7) | 0.14 |
| Acute myocardial infarction n (%) | 0 | 0 | 1.00 |
| Cardiopulmonary resuscitation n (%) | 0 | 3 (10) | 0.05 |
| Ischemic stroke n (%) | 0 | 2 (7) | 0.14 |

**Table 2. Subject characteristics and surgery comparison.**

| parameter | Without complications (n = 245) | Cardiovascular complications (n = 78) | | ANOVA $P$ |
| --- | --- | --- | --- | --- |
| | | Cardiovascular only (n = 49) | With pulmonary (n = 29) | |
| Male n (%) | 135 (55) | 26 (53) | 20 (69) | 0.33 |
| Age (years) | 65 (55–70) | 67 (63–74)* | 70 (65–73)** | <0.01 |
| BMI (kg/m$^2$) | 28.2 (24.4–31.5) | 27.4 (23.8–29.9) | 29.1 (25.3–32) | 0.64 |
| S-MPM | 4 (4–6) | 6 (4–6)** | 6 (6–6)** | <0.01 |
| Hypertension n (%) | 114 (47) | 32 (65)* | 24 (83)** | <0.01 |
| Ischemic heart disease n (%) | 17 (7) | 7 (14) | 5 (17) | 0.07 |
| COPD/asthma n (%) | 37 (15) | 7 (14)§§ | 15 (52)** | <0.01 |
| Diabetes mellitus n (%) | 31 (13) | 10 (20) | 8 (28) | 0.06 |
| Stroke n (%) | 10 (4) | 6 (12)* | 5 (17)* | <0.01 |
| **Surgery** | | | | |
| Thoracotomy n (%) | 123 (50) | 28 (57) | 22 (76)** | 0.03 |
| Lobectomy n (%) | 106 (43) | 29 (64)*§ | 24 (83)** | <0.01 |
| Bilobectomy n (%) | 6 (15) | 3 (1) | 0 | 0.23 |
| Pneumonectomy n (%) | 5 (2) | 5 (10)** | 1 (3) | 0.02 |
| Wedge resection n (%) | 128 (52) | 12 (24)** | 4 (14)** | <0.01 |
| **Post-operative Outcome** | | | | |
| Hospital LOS (days) | 6 (5–8) | 8 (7–12)**§§ | 16 (12–26)** | <0.01 |
| ICU LOS (days) | 3 (2–4) | 5 (4–7)**§§ | 10 (8–13)** | <0.01 |
| ICU readmission n (%) | 1 (0.4) | 2 (4)*§§ | 8 (28)** | <0.01 |
| 30-day mortality n (%) | 2 (1) | 0§§ | 5 (17)** | <0.01 |

* = $P<0.05$

** = $P<0.01$ vs without complications group.

§ = $P<0.05$

§§ = $P<0.01$ vs cardiovascular with pulmonary group.

BMI = body mass index; COPD = Chronic obstructive pulmonary disease; ICU = intensive care unit; LOS = length of stay; PPC = post-operative pulmonary complications; S-MPM = Surgical Mortality Probability Model.

Results of pulmonary function tests and arterial blood gases are summarized in Table 3. $FEV_1\%$, FVC%, $FEV_1/FVC\%$ and $DL_{CO}\%$ and rest $PaO_2$ were significantly lower in the cardiovascular with pulmonary complications group. $FEV_1/FVC\%$ was also lower in the cardiovascular complications only group.

Rest and exercise ventilatory parameters for all groups are summarized in Table 4. At rest, $P_{ET}CO_2$ was significantly lower in both cardiovascular complication groups. At peak exercise, $VO_2$, $VCO_2$, RER and $V_T$ were significantly lower and $f_b$ significantly higher in the cardiovascular with pulmonary complications group. Peak exercise $P_{ET}CO_2$ was lower and $V_E/VCO_2$ slope higher in both cardiovascular complication groups.

Univariate logistic regression demonstrated $V_E/VCO_2$ slope to be significantly associated with cardiovascular complications (OR = 1.10; 95%CI 1.06–1.15; $P<0.01$; AUC = 0.67) and ICU readmissions (OR = 0.92; 95% CI 0.85–0.99; $P = 0.02$; AUC = 0.71). Patients with $V_E/VCO_2$ slope ≥34.8 had significantly higher probability (17.4% vs. 46.1%) for post-operative cardiovascular complications with OR 4.1 (95% CI 2.3–7.1; $P<0.01$). However, $V_E/VCO_2$ slope association with 30-day mortality was not significant (OR = 0.93; 95%CI 0.85–1.01; $P = 0.08$; AUC = 0.68). While peak $VO_2$ was significantly associated with cardiovascular complications (OR = 0.93; 95% CI 0.89–0.98; $P<0.01$; AUC = 0.60), it was not associated with ICU

**Table 3. Pulmonary function tests and arterial blood gases comparison.**

| parameter | Without complications (n = 245) | Cardiovascular complications (n = 78) | | ANOVA P |
| --- | --- | --- | --- | --- |
| | | Cardiovascular only (n = 49) | With pulmonary (n = 29) | |
| **Pulmonary Function Tests** | | | | |
| FEV$_1$ (%predicted) | 94 ± 18 | 87 ± 19 | 82 ± 20** | <0.01 |
| FVC (%predicted) | 95 ± 16 | 92 ± 18 | 87 ± 18* | 0.02 |
| FEV$_1$/FVC (%) | 82 (76–87) | 79 (70–85)* | 77 (68–84)* | 0.01 |
| DL$_{CO}$ (% predicted) | 84 ± 21 | 79 ± 26 | 71 ± 23** | 0.01 |
| **Rest Arterial Blood Gases** | | | | |
| PaO$_2$ (mmHg) | 79 ± 9 | 76 ± 8 | 74 ± 7* | 0.01 |
| PaCO$_2$ (mmHg) | 36 (33–38) | 35 (33–38) | 35 (32–36) | 0.12 |
| pH | 7.45 (7.43–7.46) | 7.44 (7.43–7.46) | 7.44 (7.42–7.46) | 0.45 |
| **Peak Exercise Arterial Blood Gases** | | | | |
| PaO$_2$ (mmHg) | 86 (79–92) | 86 (77–92) | 78 (74–89) | 0.13 |
| PaCO$_2$ (mmHg) | 36 ± 5 | 35 ± 4 | 35 ± 5 | 0.23 |
| pH | 7.37 (7.33–7.39) | 7.36 (7.34–7.39) | 7.37 (7.33–7.41) | 0.49 |

* = $P < 0.05$

** = $P < 0.01$ vs without complications group.

§ = $P < 0.05$

§§ = $P < 0.01$ vs cardiovascular with pulmonary group.

DL$_{CO}$ = diffusing lung capacity for carbon monoxide; FEV$_1$ = forced expiratory volume in one second; FVC = forced vital capacity; PaCO$_2$ = arterial partial pressure of carbon dioxide; PaO$_2$ = arterial partial pressure of oxygen.

readmissions (OR = 1.05; 95% CI 0.95–1.16; $P$ = 0.36; AUC = 0.56) or 30-day mortality (OR = 1.11; 0.97–1.28; $P$ = 0.14; AUC = 0.61).

Multivariate logistic regression showed V$_E$/VCO$_2$ slope, age, and wedge resection, but not peak VO$_2$ to be independently associated with post-operative cardiovascular complications (Table 5) with AUC 0.75 (Fig 1). V$_E$/VCO$_2$ slope also remained significantly associated with ICU readmissions (OR = 0.92; 95% CI 0.86–0.99; $P$ = 0.03; AUC = 0.70).

Both peak VO$_2$ and V$_E$/VCO$_2$ significantly correlated with ICU LOS (rho = -0.13, $P$ = 0.02; rho = 0.38, $P < 0.01$, respectively) and hospital LOS (rho = -0.13, $P$ = 0.02; rho = 0.34, $P < 0.01$, respectively). Multiple stepwise regression analysis showed V$_E$/VCO$_2$ slope, but not peak VO$_2$ to be significantly associated with ICU LOS (b = 0.13, F = 9, $P < 0.01$) and hospital LOS (b = 0.19, F9.7, $P < 0.01$).

## Discussion

A major finding of this post-hoc analysis of a previously reported surgical cohort was that peak exercise VO$_2$ did not predict post-operative cardiovascular complications, ICU readmissions, mortality, and hospital and ICU LOS in lung resection candidates. However, in contrast V$_E$/VCO$_2$ slope predicted post-operative cardiovascular complications, ICU readmissions, and hospital and ICU LOS.

The incidence of post-operative cardiovascular complications was 22%, similar to previous reports [17,18]. The most frequent cardiovascular complication was supraventricular arrhythmia (60%) in agreement with previous studies of post-lung resection surgery [18,19]. The second most common complication was post-operative hypotension (58%), which is also frequent [20]. Although these complications may be considered mild, post-operative

**Table 4. Cardiopulmonary exercise testing comparison.**

| parameter | Without complications (n = 245) | Cardiovascular complications (n = 78) | | ANOVA *P* |
|---|---|---|---|---|
| | | Cardiovascular only only (n = 49) | With pulmonary (n = 29) | |
| **Rest Ventilation and Gas Exchange** | | | | |
| $VO_2$ (ml/kg/min) | 4.2 (3.5–5.0) | 4.2 (3.1–4.6) | 4.1 (3.4–5.2) | 0.51 |
| $VCO_2$ (ml/min) | 0.25 ± 0.10 | 0.22 ± 0.08 | 0.22 ± 0.10 | 0.04 |
| $V_E$ (l/min) | 10 ± 4 | 9 ± 3 | 10 ± 5 | 0.47 |
| $V_T$ (ml) | 0.57 (0.40–0.74) | 0.57 (0.43–0.71) | 0.45 (0.31–0.65) | 0.19 |
| $f_b$ (bpm) | 18 (15–21) | 17 (14–21)§§ | 20 (18–23)* | 0.01 |
| $V_D/V_T$ | 0.29 (0.23–0.34) | 0.30 (0.24–0.34) | 0.29 (0.23–0.34) | 0.77 |
| $P_{ET}CO_2$ (mmHg) | 30 (27–32) | 28 (24–31)*§ | 26 (23–28)** | <0.01 |
| **Peak Exercise Ventilation and Gas Exchange** | | | | |
| $VO_2$ (ml/kg/min) | 19.9 (16.5–25) | 19.0 (16–23.1) | 16.3 (15–20.3)** | 0.01 |
| $VCO_2$ (ml/min) | 1.67 (1.30–2.10) | 1.56 (1.23–1.95) | 1.32 (1.03–1.71)** | <0.01 |
| RER | 1.05 (0.92–1.14) | 0.99 (0.88–1.12) | 0.97 (0.82–1.05)** | 0.01 |
| $V_E$ (l/min) | 54 (44–68) | 53 (41–65) | 48 (43–61) | 0.31 |
| $V_T$ (ml) | 1.75 (1.37–2.11) | 1.74 (1.27–2.14) | 1.42 (1.08–1.85)** | 0.01 |
| $f_b$ (bpm) | 32 (28–36) | 33 (30–37) | 35 (32–41)** | 0.01 |
| $V_D/V_T$ | 0.21 ± 0.07 | 0.22 ± 0.06 | 0.24 ± 0.06 | 0.12 |
| $P_{ET}CO_2$ (mmHg) | 36 ± 5 | 33 ± 6*§ | 30 ± 5** | <0.01 |
| $V_E/VCO_2$ slope | 29 (25–33) | 31 (27–37)*§§ | 37 (34–42)** | <0.01 |

* = *P*<0.05

** = *P*<0.01 vs without complications group.

§ = *P*<0.05

§§ = *P*<0.01 vs cardiovascular with pulmonary group.

$f_b$ = breathing frequency; $P_{ET}CO_2$ = partial pressure of end-tidal carbon dioxide; PPC = post-operative pulmonary complications; RER = respiratory exchange ratio; $VCO_2$ = carbon dioxide output; $V_D$ = dead space volume; $V_E$ = minute ventilation; $V_E/VCO_2$ = ventilatory efficiency; $VO_2$ = oxygen consumption; $V_T$ = tidal volume.

arrhythmias are associated with longer hospital LOS, intensive care unit admissions, resource utilization and costs of care [17,21] and post-operative hypotension may be associated with myocardial injury [20]. Indeed, in our cohort, ICU readmissions were greater and hospital and

**Table 5. Logistic regression analysis.**

| | univariate | | | multivariate stepwise | | |
|---|---|---|---|---|---|---|
| | OR | 95 CI | p | OR | 95 CI | *P* |
| Age | 1.05 | 1.02–1.08 | <0.01 | 1.04 | 1.01–1.07 | 0.02 |
| S-MPM | 1.63 | 1.29–2.07 | <0.01 | NS | | |
| Thoracotomy No (%) | 1.77 | 1.05–3.00 | 0.03 | NS | | |
| Lobectomy No (%) | 2.78 | 1.62–4.76 | <0.01 | NS | | |
| Wedge resection No (%) | 0.24 | 0.13–0.43 | <0.01 | 0.28 | 0.15–0.53 | <0.01 |
| $FEV_1/FVC$ | 0.97 | 0.94–0.99 | <0.01 | NS | | |
| $DL_{CO}$ | 0.98 | 0.97–1.00 | 0.01 | NS | | |
| peak $VO_2$ | 0.93 | 0.89–0.98 | <0.01 | NS | | |
| $V_E/VCO_2$ slope | 1.10 | 1.06–1.15 | <0.01 | 1.06 | 1.01–1.11 | 0.01 |

$DL_{CO}$ = diffusing lung capacity for carbon monoxide; $FEV_1$ = forced expiratory volume in one second; FVC = forced vital capacity; $P_{ET}CO_2$ = partial pressure of end-tidal carbon dioxide; S-MPM = Surgical Mortality Probability Model; $V_E/VCO_2$ = ventilatory efficiency; $VO_2$ = oxygen consumption.

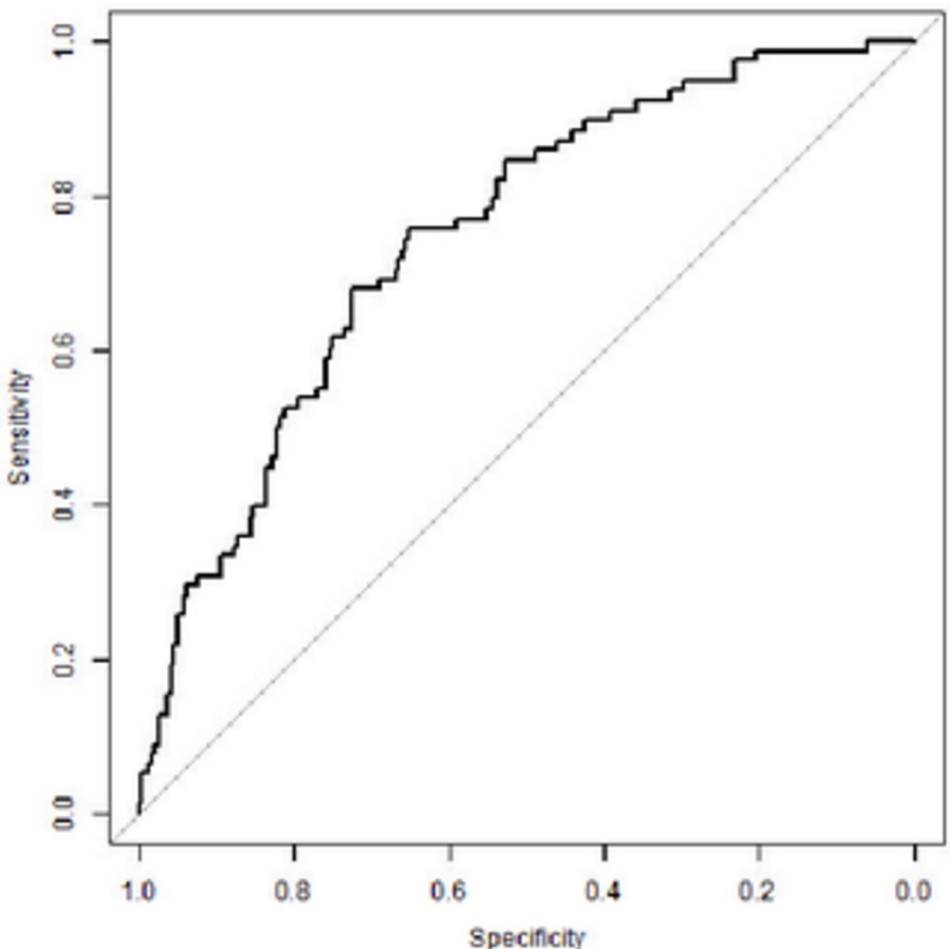

**Fig 1. Receiver operating characteristic curve of $V_E/VCO_2$ slope and post-operative cardiovascular complications.**
Multivariate logistic regression model [AUC = 0.747 (0.687; 0.807)].

ICU LOS were significantly longer in both cardiovascular complication groups. Severe complications (except of hypotension and arrhythmia) were relatively rare (16%) and developed in only 10 patients.

Compared to patients without post-operative complications, patients who developed cardiovascular complications (with or without pulmonary complications) were significantly older, had more comorbidities and lower pre-operative $FEV_1/FVC\%$. Lobectomy was more frequent and wedge resection less frequent in these patients, in concordance with previous studies [17,18]. In our study, patients with a combination of cardiovascular and pulmonary complications had previously established risk factors for post-operative pulmonary complications including thoracotomy, lower $FEV_1\%$, FVC%, $DL_{CO}\%$ and $PaO_2$ [22,23]. The 30-day mortality was also higher in this group of patients.

During pre-operative exercise, patients with post-operative cardiovascular complications (with or without pulmonary complications) exhibited significantly higher $V_E/VCO_2$ slope. In contrast, peak $VO_2$ was significantly lower only in the cardiovascular with pulmonary complications group. $V_E/VCO_2$ slope is determined by ventilatory control impairment [6] suggesting it may be a parameter which is predictive of post-operative pulmonary complications. Indeed, several studies have previously shown its superiority to peak $VO_2$ [4,5,10,14,24,25]. However,

the main determinant of peak $VO_2$ is cardiac output [7], which has been shown to be a major risk factor for post-operative cardiovascular complications [9].

Peak $VO_2$ is considered a parameter for assessment of cardiovascular fitness [8]. Hence, an association with cardiovascular complications might be expected. However, in this study, although the peak $VO_2$ was significantly associated with cardiovascular complications by univariate analysis, multivariate logistic regression showed that only $V_E/VCO_2$ slope, type of surgical procedure and age were independently associated with cardiovascular complications. This is in agreement with prior studies in chronic heart failure patients where $V_E/VCO_2$ slope has also been found superior to peak $VO_2$ in the prediction of cardiovascular events [26–28]. The observed association of $V_E/VCO_2$ slope and cardiovascular complications might be explained by an inverse association of $V_E/VCO_2$ slope with cardiac output and positive association with capillary wedge pressure [29]. Moreover, unlike peak $VO_2$, $V_E/VCO_2$ slope is independent of patient effort and peripheral skeletal muscle function enabling acquisition during submaximal exercise ($V_E/VCO_2$ has been validated across different RER [30]) which may be beneficial for assessment of functionally limited lung surgery candidates.

Our findings may have clinical implications. First, peak $VO_2$ is part of risk stratification algorithms in both American and European guidelines [1,2]. However, multiple studies have previously shown its poor prognostic utility [3–5,10]. In contrast, $V_E/VCO_2$ slope has been shown to be significantly better in the prediction of pulmonary [4,5,10,14,24,25] and now also cardiovascular complications. Accordingly, we suggest that lung resection candidate risk stratification algorithms consider routine inclusion of $V_E/VCO_2$ slope. Second, pre-operative risk stratification may allow pre-operative patient optimization as $V_E/VCO_2$ slope may be a therapeutic target [31] and promote better perioperative planning [17].

This study has several limitations. First, according to published ERS/ESTS guidelines [2], subjects with peak predicted post-operative $VO_2<10$ml/kg/min were excluded, which may diminish the predictive value of peak $VO_2$. However, only 6 subjects from our study cohort were excluded for this reason. Second, chronic medication data were unavailable. However, chronic medication is usually associated with the underlying diseases and our logistic regression analysis was adjusted for those. Third, $PaO_2$ was significantly lower in patients with cardiovascular and pulmonary complications at rest. As $VO_2$ is determined not only by cardiac output but also arteriovenous oxygen difference, this may have contributed to the observed $VO_2$ differences (although these were observed only at peak exercise). Therefore, we cannot exclude, that peak $VO_2$ may be a relevant prognostic factor in this subgroup of severely ill patients.

## Conclusion

Peak $VO_2$ is a poor predictor of post-operative cardiovascular complications. In contrast, the $V_E/VCO_2$ slope predicted post-operative cardiovascular complications, ICU readmissions and both hospital and ICU LOS. Our observations suggest consideration of routine inclusion of $V_E/VCO_2$ slope for pre-operative risk stratification in lung resection candidates.

## Supporting information

**S1 Table. Comparison of patients with and without cardiovascular complications.** Comparison of patients with and without cardiovascular complications.
(DOCX)

**S2 Table. Dataset.** Minimal dataset.
(XLSX)

## Author Contributions

**Conceptualization:** Andrej Mazur, Kristian Brat, Pavel Homolka, Michal Svoboda, Monika Bratova, Vladimir Sramek, Lyle J. Olson, Ivan Cundrle.

**Data curation:** Andrej Mazur, Kristian Brat, Pavel Homolka, Zdenek Merta, Michal Svoboda, Monika Bratova, Vladimir Sramek, Lyle J. Olson, Ivan Cundrle.

**Formal analysis:** Michal Svoboda, Ivan Cundrle.

**Funding acquisition:** Kristian Brat, Ivan Cundrle.

**Investigation:** Andrej Mazur, Kristian Brat, Pavel Homolka, Zdenek Merta, Monika Bratova, Vladimir Sramek, Ivan Cundrle.

**Methodology:** Andrej Mazur, Kristian Brat, Pavel Homolka, Zdenek Merta, Michal Svoboda, Vladimir Sramek, Lyle J. Olson, Ivan Cundrle.

**Project administration:** Ivan Cundrle.

**Resources:** Kristian Brat, Ivan Cundrle.

**Supervision:** Kristian Brat, Vladimir Sramek, Lyle J. Olson, Ivan Cundrle.

**Validation:** Kristian Brat, Zdenek Merta, Michal Svoboda, Monika Bratova, Lyle J. Olson, Ivan Cundrle.

**Writing – original draft:** Andrej Mazur, Kristian Brat, Pavel Homolka, Zdenek Merta, Michal Svoboda, Monika Bratova, Vladimir Sramek, Lyle J. Olson, Ivan Cundrle.

**Writing – review & editing:** Andrej Mazur, Kristian Brat, Pavel Homolka, Zdenek Merta, Michal Svoboda, Monika Bratova, Vladimir Sramek, Lyle J. Olson, Ivan Cundrle.

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
