## [Decision Letter · Decision Letter 0]

16 Jun 2022

PONE-D-22-05505Ventilatory efficiency is superior to peak oxygen uptake for prediction of lung resection cardiovascular complicationsPLOS ONE

Dear Dr. Cundrle Jr.,

Thank you for submitting your manuscript to PLOS ONE. After careful consideration, we feel that it has merit but does not fully meet PLOS ONE’s publication criteria as it currently stands. Therefore, we invite you to submit a revised version of the manuscript that addresses the points raised during the review process.

Your manuscript has been assessed by two expert reviewers, whose comments are appended below. As you will see from the full reports, the reviewers have highlighted a number of concerns regarding the study design, reporting of the methodology and other organizational points. Please ensure you respond to each point carefully in your response to reviewers document, and modify your manuscript accordingly.

We look forward to receiving your revised manuscript.

Kind regards,

Joseph Donlan

Editorial Office

PLOS ONE

Journal Requirements:

Reviewers' comments:

Reviewer's Responses to Questions

**Comments to the Author**

1. Is the manuscript technically sound, and do the data support the conclusions?

Reviewer #1: Yes

Reviewer #2: Partly

2. Has the statistical analysis been performed appropriately and rigorously? 

Reviewer #1: Yes

Reviewer #2: No

3. Have the authors made all data underlying the findings in their manuscript fully available?

Reviewer #1: Yes

Reviewer #2: Yes

4. Is the manuscript presented in an intelligible fashion and written in standard English?

Reviewer #1: Yes

Reviewer #2: Yes

5. Review Comments to the Author

Reviewer #1: Congratulations, you made a great job! Your study is very important for clinical practice and scientific knowledge.

However, some corrections must be taking in count:

ABSTRACT

introduction -> The first sentence is almost a conclusion sentence. I suggest you rewrite or exclude.

methods -> You should mention the evaluation method (Subjects underwent preoperative CPET). You mentioned Friedman’s test in the abstract, but not in the body text.

INTRODUCTION

The first two paragraphs are very precise, however the introduction is too short. Please include one more paragraph.

The aim of the study do no match the title, discussion and conclusions. Please rewrite the aim. Your results and discussions are very clear: Ventilatory efficiency is superior to peak oxygen uptake for prediction of lung resection cardiovascular complications. However, your aim was "to compare pre-operative cardiopulmonary exercise testing parameters in patients who developed post-operative cardiovascular complications with patients who did not after lung resection surgery". Actually you did it, however your discussion and conclusion were made based in two variables (VO2 and VE/VCO2 slope) for complications prediction. In my opinion, the aim of your study was to analyze/compare these two variables for complications prediction.

METHODS

Include a "Study desing" topic. You should include in this new topic the preoperative evaluation method (Subjects underwent preoperative CPET) and some sentences/paragraphs should be moved from "subject selection" to study desing topic:

"This is a post-hoc analysis of a previously published prospective multicenter study which evaluated pre-operative rest ventilatory parameters as predictors of post-operative pulmonary complications in lung resection candidates" -> lines 67-69

"The study was conducted in accordance with the declaration of Helsinki. All participants provided written informed consent and the study was approved by the local Ethics Committee of St. Anne’s University Hospital in Brno (reference No. 19JS/2017, date of approval April 12, 2017; reference No. 2G/2018, date of approval March 21st, 2018) and by the local Ethics Committee of the University Hospital Brno (reference No. 150617/EK, date of approval June 19th, 2017). The study was registered at ClinicalTrials.gov (NCT03498352) and the manuscript adheres to the applicable STROBE guidelines." -> lines 76-82.

You mentioned Friedman’s test in the abstract, but not in the body text methods.

RESULTS

9% of the patients were excluded due to pulmonary complications only, but it was not an exclusion criteria. Please add "pulmonary complications only" as an exclusion criteria.

You mentioned the abbreviation S-MPM (line 143, 144) in the text, but only described it in the table.

DISCUSSION

These data must be in results, not in discussion -> "Patients with VE/VCO2 slope ≥34.8 had significantly higher probability (17.4% vs. 46.1%) for post-operative cardiovascular complications with OR 4.1 (95% CI 2.3-7.1; P<0.01)."

CONCLUSION

Remove the term "In conclusion" at the beginning of the sentence.

Be more specific -> Our observations suggest consideration of routine inclusion of VE/VCO2 slope for pre-operative risk stratification in candidates for LUNG RESECTION.

Reviewer #2: Comments:

First, I would like to congratulate the authors for the research. Post-operative prediction of complications is a useful clinical application of cardiopulmonary exercise testing (CPET).

Regarding the manuscript, I have some considerations:

1. Abstract:

Line 24 (and in Introduction - line 57) the authors stated that “VO2 is related to cardiac output.”

The statement is correct, but incomplete. We must not forget that according to “FICK principle”, oxygen uptake (VO2) is the product of cardiac output and arteriovenous oxygen difference. The importance of arteriovenous oxygen difference cannot be neglected, mainly in patients with pulmonary diseases, which can have hypoxia at rest and during exercise. It’s important to consider that this alteration can impact on VO2 by reducing arterial oxygen content:

Arterial oxygen content = (Hemoglobin x 1.36 x SaO2) + (0.0031 x PaO2).

Then, diseases that can affect arterial oxygen, can also have consequences in arteriovenous oxygen difference and VO2 as well.

2. Introduction:

Lines 60-63: the authors stated that “aim of this study was to compare pre-operative cardiopulmonary exercise testing parameters in patients who developed post-operative cardiovascular complications with patients who did not after lung resection surgery.”

The aim of the study was well stablished. That’s why I did not clearly understood why cardiovascular complications were reported only subdivided in “with” or “without pulmonary complications”. This division is interesting and could have been made, but the overall data for cardiovascular complications should have been reported. Subsequent division would be a subgroup analysis.

3. Methods:

Lines 101-111: The definition of cardiovascular complications were wide, ranging from auto-limited arrhythmias and hypotension to cardiopulmonary resuscitation. Conversely, definition of pulmonary complications was clinically narrower.

Probably this was due to the study design that was focused on Pulmonary complications as a Primary Outcome. Cardiovascular Outcome was previously defined as a Secondary Outcome, as described in clinical trial registry (NCT03498352 - Rest Ventilatory Parameters Predict Morbidity and Mortality in Thoracic Surgery).

Thus, a narrowing of the cardiovascular complication or a creating subdivision in minor versus major complication would have been more informative.

4) Results. Table 1 (page 8) describes the type and frequency of the cardiovascular complications.

Most of the complications were related to arrhythmias and hypotension.

Only 3 of the 49 complications (6%) in the “Cardiovascular only subgroup” were not related to arrhythmias and hypotension, while 10 of 29 complication (34%) in the “With pulmonary subgroup” were not related to arrhythmias and hypotension. Thus, cardiovascular complications in the “with pulmonary subgroup” were clinically more relevant.

Thus, the subdivision of the cardiovascular complications generated heterogeneity of the types of complications in the subgroups, and uncertainty to the analysis, as it can influence the prediction value of CPET.

Including the overall analysis could provide more information of the cardiovascular complications and it’s relation with CPET, reducing the uncertainty generated by the analysis of only the subgroups.

5. Results. Table 2 (pages 8-9) describes subjects characteristic.

Patients with complications were older, and there were differences in the type of surgery and post-operative outcomes.

Regarding subgroups of cardiovascular complications, “with pulmonary subgroup” were older (70 versus 67 years), had a higher BMI (29.1 versus 27.4) and heterogeneity in types of the surgery.

Cardiovascular risk factors and medication use were not reported in the subjects’ characteristics.

Thus, there are several clinical characteristics (reported and non-reported) that could have been related to a higher risk profile in the Cardiovascular with pulmonary complication subgroup and could have reduced the prediction value of CPET.

6. Results. Table 3. Pulmonary function (Page 10-11).

Rest pulmonary function were different among the three subgroups.

PaO2 at rest and at peak exercise were lower in the “with pulmonary” subgroup. Maybe that’s why CPET were clinically different in this subgroup, with significantly lower peak VO2.

PaO2 is related to arterial oxygen content and, consequently, arteriovenous oxygen difference and VO2 (FICK Principle).

Hence, along with all other clinical characteristics, peak VO2 might be a predictor within this subgroup with more relevant cardiovascular complications (the difference is not only the presence of pulmonary complications, as discussed above). The suggestion is to include some considerations in the discussion.

7. Logistic regression analysis (page 13)

There were several variables associated to cardiovascular complication in the univariate model, including age, type of surgery, rest pulmonary functions and CPET.

In multivariate stepwise, only age, wedge resection ant VE/VCO2 slope were significant. It must be considered that age and wedge resection were different in the subgroups of cardiovascular complications (with or without pulmonary).

Wedge resection was the most important factor (OR 0.28). Effect size for VE/VCO2 slope (OR 1.06) and age (1.04) were of much smaller magnitude.

Hence, the type of surgery (Wedge resection) could have reduced the effect size for peak VO2 and maybe that’s why it was not significant in multivariate stepwise (In univariate, the OR was 0.93).

Also, there was a wide definition for cardiovascular complication, and this could have affected the predictive model ability to detect relevant impact of CPET variables.

Suggestions

- Review the logistic regression model, as there may be non-reported clinical characteristics that were not analysed (hypertension, diabetes, medication use).

- The major influence for cardiovascular complication risk was the type of surgery (Wedge resection). A reanalysis of logistic regression model removing these patients, may be interesting to increase the sensibility of the CPET predictive variables in the other types of surgeries.

- Cardiovascular complication definition was too wide. Maybe a subgroup analysis with subdivision in minor versus major cardiovascular complications maybe more clinically relevant.

6. PLOS authors have the option to publish the peer review history of their article (what does this mean?). If published, this will include your full peer review and any attached files.

Reviewer #1: **Yes: **Murillo Frazão

Reviewer #2: No

---

## [Author Response · Author response to Decision Letter 0]

7 Jul 2022

Reviewer #1: 

ABSTRACT

introduction -> The first sentence is almost a conclusion sentence. I suggest you rewrite or exclude.

methods -> You should mention the evaluation method (Subjects underwent preoperative CPET). You mentioned Friedman’s test in the abstract, but not in the body text.

We would like to thank the reviewer for these suggestions. We have slightly rewritten the first sentence, added CPET in the methods and removed Friedman’s test. 

“Ventilatory efficiency (VE/VCO2 slope) has been shown superior to peak oxygen consumption (VO2) for prediction of post-operative pulmonary complications in patients undergoing thoracotomy.” Line 21-23

“All of the patients underwent preoperative cardiopulmonary exercise testing.” Line 28

“One-way analysis of variance or the Kruskal–Wallis test, and multivariate logistic regression were used for statistical analysis and data summarized as median (IQR).” Line 30-31

INTRODUCTION

The first two paragraphs are very precise, however the introduction is too short. Please include one more paragraph.

We would like to thank the reviewer for this suggestion; we have added one more paragraph in the introduction as follows: 

“In contrast, determinants of VO2 include cardiac output (7) and arteriovenous oxygen difference. Peak VO2 is considered a cardiovascular fitness parameter (8) and was shown to be a major risk factor for post-operative cardiovascular complications (9). Hence, an association with cardiovascular complications might be expected.” Line 57-60

The aim of the study do no match the title, discussion and conclusions. Please rewrite the aim. Your results and discussions are very clear: Ventilatory efficiency is superior to peak oxygen uptake for prediction of lung resection cardiovascular complications. However, your aim was "to compare pre-operative cardiopulmonary exercise testing parameters in patients who developed post-operative cardiovascular complications with patients who did not after lung resection surgery". Actually you did it, however your discussion and conclusion were made based in two variables (VO2 and VE/VCO2 slope) for complications prediction. In my opinion, the aim of your study was to analyze/compare these two variables for complications prediction.

We would like to thank the reviewer for this comment. We agree and have rewritten the aim of the study as follows.

“Accordingly, the aim of this study was to compare pre-operative peak exercise VO2 and VE/VCO2 slope in the prediction of post-operative cardiovascular complications in patients after lung resection surgery.” Line 62-64

METHODS

Include a "Study desing" topic. You should include in this new topic the preoperative evaluation method (Subjects underwent preoperative CPET) and some sentences/paragraphs should be moved from "subject selection" to study desing topic:

"This is a post-hoc analysis of a previously published prospective multicenter study which evaluated pre-operative rest ventilatory parameters as predictors of post-operative pulmonary complications in lung resection candidates" -> lines 67-69

"The study was conducted in accordance with the declaration of Helsinki. All participants provided written informed consent and the study was approved by the local Ethics Committee of St. Anne’s University Hospital in Brno (reference No. 19JS/2017, date of approval April 12, 2017; reference No. 2G/2018, date of approval March 21st, 2018) and by the local Ethics Committee of the University Hospital Brno (reference No. 150617/EK, date of approval June 19th, 2017). The study was registered at ClinicalTrials.gov (NCT03498352) and the manuscript adheres to the applicable STROBE guidelines." -> lines 76-82.

We would like to thank the reviewer for this comment. We have added a Study design paragraph as suggested. 

“Study design

This is a post-hoc analysis of a previously published prospective multicenter study which evaluated pre-operative rest ventilatory parameters as predictors of post-operative pulmonary complications in lung resection candidates (4). All of the included patients underwent preoperative CPET and pulmonary function tests. Post-operative cardiovascular complications were prospectively accessed from the first 30 post-operative days or from the hospital stay. The study was conducted in accordance with the declaration of Helsinki. All participants provided written informed consent and the study was approved by the local Ethics Committee of St. Anne’s University Hospital in Brno (reference No. 19JS/2017, date of approval April 12, 2017; reference No. 2G/2018, date of approval March 21st, 2018) and by the local Ethics Committee of the University Hospital Brno (reference No. 150617/EK, date of approval June 19th, 2017). The study was registered at ClinicalTrials.gov (NCT03498352) and the manuscript adheres to the applicable STROBE guidelines.” Line 75-87

You mentioned Friedman’s test in the abstract, but not in the body text methods.

We would like to than the reviewer for pointing this out. We have removed Friedman’s test from the abstract. 

“One-way analysis of variance or the Kruskal–Wallis test, and multivariate logistic regression were used for statistical analysis and data summarized as median (IQR).” Line 30-31

RESULTS

9% of the patients were excluded due to pulmonary complications only, but it was not an exclusion criteria. Please add "pulmonary complications only" as an exclusion criteria.

We would like to thank the reviewer for this comment. We have modified exclusion criteria. 

“Exclusion criteria were contraindication for lung resection because of inoperability or low peak predicted post-operative VO2 [VO2<10ml/kg/min or <35% predicted together with predicted post-operative forced expiratory volume at one second (FEV1) <30% and diffusing capacity for carbon monoxide (DLCO) <30% 2)] and development of pulmonary complications only.” Line 70-74

You mentioned the abbreviation S-MPM (line 143, 144) in the text, but only described it in the table.

We would like to thank the reviewer for pointing this out. We have defined this abbreviation in the text. 

“Surgical Mortality Probability Model (S-MPM), an ASA (American Society of Anesthesiologists) physical status adjusted for surgery risk class and emergency status (15), was used to allow patient’s co-morbidities comparison.” Line 121-124

DISCUSSION

These data must be in results, not in discussion -> "Patients with VE/VCO2 slope ≥34.8 had significantly higher probability (17.4% vs. 46.1%) for post-operative cardiovascular complications with OR 4.1 (95% CI 2.3-7.1; P<0.01)."

We would like to thank the reviewer for this comment. We have moved the data to the results section. 

“Patients with VE/VCO2 slope ≥34.8 had significantly higher probability (17.4% vs. 46.1%) for post-operative cardiovascular complications with OR 4.1 (95% CI 2.3-7.1; P<0.01).” Line 198-199

CONCLUSION

Remove the term "In conclusion" at the beginning of the sentence.

Be more specific -> Our observations suggest consideration of routine inclusion of VE/VCO2 slope for pre-operative risk stratification in candidates for LUNG RESECTION.

We would like to thank the reviewer for this suggestion. We have rewritten the conclusion. 

“Peak VO2 is a poor predictor of post-operative cardiovascular complications. In contrast, the VE/VCO2 slope predicted post-operative cardiovascular complications, ICU readmissions and both hospital and ICU LOS. Our observations suggest consideration of routine inclusion of VE/VCO2 slope for pre-operative risk stratification in lung resection candidates.“ Line 295-299

Reviewer #2: 

1. Abstract:

Line 24 (and in Introduction - line 57) the authors stated that “VO2 is related to cardiac output.”

The statement is correct, but incomplete. We must not forget that according to “FICK principle”, oxygen uptake (VO2) is the product of cardiac output and arteriovenous oxygen difference. The importance of arteriovenous oxygen difference cannot be neglected, mainly in patients with pulmonary diseases, which can have hypoxia at rest and during exercise. It’s important to consider that this alteration can impact on VO2 by reducing arterial oxygen content:

Arterial oxygen content = (Hemoglobin x 1.36 x SaO2) + (0.0031 x PaO2).

Then, diseases that can affect arterial oxygen, can also have consequences in arteriovenous oxygen difference and VO2 as well.

We would like to thank the reviewer for this comment. We have added arteriovenous oxygen difference in both abstract and introduction. 

“VE/VCO2 slope is determined by ventilatory drive and ventilation/perfusion mismatch whereas VO2 is related to cardiac output and arteriovenous oxygen difference.” Line 23-25

“In contrast, determinants of VO2 include cardiac output (7) and arteriovenous oxygen difference.” Line 57-58

2. Introduction:

Lines 60-63: the authors stated that “aim of this study was to compare pre-operative cardiopulmonary exercise testing parameters in patients who developed post-operative cardiovascular complications with patients who did not after lung resection surgery.”

The aim of the study was well stablished. That’s why I did not clearly understood why cardiovascular complications were reported only subdivided in “with” or “without pulmonary complications”. This division is interesting and could have been made, but the overall data for cardiovascular complications should have been reported. Subsequent division would be a subgroup analysis.

We would like to thank the reviewer for this comment. Division to subgroups was done because cardiovascular and pulmonary complications often coexist, and despite prospective assessment of complications, we were not able to fully distinguish which complications were primary and which were secondary. To mitigate this, we divided our cardiovascular complications subjects into 2 groups – cardiovascular only and cardiovascular with pulmonary complications (frequency of cardiovascular complications was nearly the same in both groups Table 1) and analyzed them separately. We commented on this in the manuscript. Moreover, we have now included a Supplement Table 1 comparing no complications vs. cardiovascular complications group (both groups together). Please see Supplement Table 1. We have also modified first paragraph of the results section as follows.

“Thirty patients (9%) developed pulmonary complications only and were excluded; 78 patients (22%) developed cardiovascular complications and 245 patients (69%) had no complications. Comparison of patients with and without cardiovascular complications is shown in Supplement Table 1. Out of the 78 patients with cardiovascular complications, 49 patients developed cardiovascular complications only and 29 patients developed cardiovascular with pulmonary complications. Cardiovascular and pulmonary complications often coexist (16), and despite prospective assessment of complications, we were not able to fully distinguish which complications were primary and which were secondary. To mitigate this, we divided our cardiovascular complications subjects into 2 groups – cardiovascular only and cardiovascular with pulmonary complications (frequency of cardiovascular complications was nearly the same in both groups Table 1) and analyzed them separately.” Line 145-155

3. Methods:

Lines 101-111: The definition of cardiovascular complications were wide, ranging from auto-limited arrhythmias and hypotension to cardiopulmonary resuscitation. Conversely, definition of pulmonary complications was clinically narrower.

Probably this was due to the study design that was focused on Pulmonary complications as a Primary Outcome. Cardiovascular Outcome was previously defined as a Secondary Outcome, as described in clinical trial registry (NCT03498352 - Rest Ventilatory Parameters Predict Morbidity and Mortality in Thoracic Surgery).

Thus, a narrowing of the cardiovascular complication or a creating subdivision in minor versus major complication would have been more informative.

We would like to thank the reviewer for this comment and we understand these concerns. Cardiovascular complications were defined as previously recommended (Bennett-Guerrero E, Welsby I, Dunn TJ, Young LR, Wahl TA, Diers TL, et al. The Use of a Postoperative Morbidity Survey to Evaluate Patients with Prolonged Hospitalization After Routine, Moderate-Risk, Elective Surgery. Anesthesia & Analgesia. 1999 Aug;89(2):514–9.) We agree with the reviewer, that dividing complications into minor and major would be interesting. However, the number of major complications (including pulmonary edema, pulmonary embolism, heart failure, acute myocardial infarction, cardiopulmonary resuscitation and stroke) was low 13 (16%) and presented “only” in 10 patients (13%). Comparison of such a small group would be hard / impossible to interpret. We hope the reviewer will agree. We comment on this in the manuscript.

“The incidence of post-operative cardiovascular complications was 22%, similar to previous reports (17,18). The most frequent cardiovascular complication was supraventricular arrhythmia (60%) in agreement with previous studies of post-lung resection surgery (18,19). The second most common complication was post-operative hypotension (58%), which is also frequent (20). Although these complications may be considered mild, post-operative arrhythmias are associated with longer hospital LOS, intensive care unit admissions, resource utilization and costs of care (17,21) and post-operative hypotension may be associated with myocardial injury (20). Indeed, in our cohort, ICU readmissions were greater and hospital and ICU LOS were significantly longer in both cardiovascular complication groups. Severe complications (except of hypotension and arrhythmia) were relatively rare (16%) and developed in only 10 patients.“ Line 230-240

4) Results. Table 1 (page 8) describes the type and frequency of the cardiovascular complications.

Most of the complications were related to arrhythmias and hypotension.

Only 3 of the 49 complications (6%) in the “Cardiovascular only subgroup” were not related to arrhythmias and hypotension, while 10 of 29 complication (34%) in the “With pulmonary subgroup” were not related to arrhythmias and hypotension. Thus, cardiovascular complications in the “with pulmonary subgroup” were clinically more relevant.

Thus, the subdivision of the cardiovascular complications generated heterogeneity of the types of complications in the subgroups, and uncertainty to the analysis, as it can influence the prediction value of CPET.

Including the overall analysis could provide more information of the cardiovascular complications and it’s relation with CPET, reducing the uncertainty generated by the analysis of only the subgroups.

We would like to thank the reviewer for this comment and we completely agree. However, it may be a misunderstanding. The most important part, the univariate and multivariate logistic regression, was done irrespective of the subdivision. Moreover, as suggested by the reviewer, we have now included a Supplement Table 1 comparing no complications vs. cardiovascular complications group (both groups together). Please see Supplement Table 1.

5. Results. Table 2 (pages 8-9) describes subjects characteristic.

Patients with complications were older, and there were differences in the type of surgery and post-operative outcomes.

Regarding subgroups of cardiovascular complications, “with pulmonary subgroup” were older (70 versus 67 years), had a higher BMI (29.1 versus 27.4) and heterogeneity in types of the surgery.

Cardiovascular risk factors and medication use were not reported in the subjects’ characteristics.

Thus, there are several clinical characteristics (reported and non-reported) that could have been related to a higher risk profile in the Cardiovascular with pulmonary complication subgroup and could have reduced the prediction value of CPET.

We would like to thank the reviewer for this comment. All of the potential confounders (parameters significantly different between groups, and without significant in-between correlation) were included in the multivariate logistic regression model (including age, type of surgery and comorbidities summarized as the S-MPM; please note BMI was not significantly different between groups, and is not different even in the newly included overall (2 groups) comparison - Supplement Table 1). Please see Table 2, Supplement table 1. We comment on logistic regression parameters selection in the statistics section. 

“Univariate logistic regression was done for parameters significantly different between groups. The Spearman rank test was performed to evaluate parameters with significant correlation. Parameters with correlation coefficients <±0.5 were used for multivariate stepwise logistic and linear regression.” Line 129-132

In the presented study, cardiovascular risk factors were part of the Surgical Mortality Probability Model (S-MPM), which is basically an ASA physical status adjusted for surgery risk and emergency status. Using this parameter allowed us to control the logistic regression for such a complex confounder as are comorbidities (and not creating an over-fitting model). For the reviewer, we have included frequency of the most common comorbidities in Table 2 (and also in the Supplement Table 1). As you can see, the frequency of comorbidities practically mirrors the S-MPM. We have also added a detailed explanation of components of S-MPM in the methods section.

“Surgical Mortality Probability Model (S-MPM), an ASA (American Society of Anesthesiologists) physical status adjusted for surgery risk class and emergency status (15), was used to allow patient’s co-morbidities comparison.” Line 121-124

We agree with the reviewer, that information about medication would be useful. Unfortunately, we do not have this data readily available. Therefore, we have decided to include this in the limitation section. Moreover, chronic medication is usually bound to the associated disease and our analyses were adjusted for those. We hope the reviewer will agree. 

“Second, chronic medication data were unavailable. However, chronic medication is usually associated with the underlying diseases and our logistic regression analysis was adjusted for those.” Line 284-286

6. Results. Table 3. Pulmonary function (Page 10-11).

Rest pulmonary function were different among the three subgroups.

PaO2 at rest and at peak exercise were lower in the “with pulmonary” subgroup. Maybe that’s why CPET were clinically different in this subgroup, with significantly lower peak VO2.

PaO2 is related to arterial oxygen content and, consequently, arteriovenous oxygen difference and VO2 (FICK Principle).

Hence, along with all other clinical characteristics, peak VO2 might be a predictor within this subgroup with more relevant cardiovascular complications (the difference is not only the presence of pulmonary complications, as discussed above). The suggestion is to include some considerations in the discussion.

We would like to thank the reviewer for this suggestion. We have added a statement in the limitation section. Please note, PaO2 was significantly lower only at rest in the cardiovascular with pulmonary complications group. Whereas VO2 was significantly lower at peak exercise, but not at rest. However, we understand the reviewers point and added a statement in the limitations section. 

“Third, PaO2 was significantly lower in patients with cardiovascular and pulmonary complications at rest. As VO2 is determined not only by cardiac output but also arteriovenous oxygen difference, this may have contributed to the observed VO2 differences (although these were observed only at peak exercise). Therefore, we cannot exclude, that peak VO2 may be a relevant prognostic factor in this subgroup of severely ill patients. ” Line 286-291

7. Logistic regression analysis (page 13)

There were several variables associated to cardiovascular complication in the univariate model, including age, type of surgery, rest pulmonary functions and CPET.

In multivariate stepwise, only age, wedge resection ant VE/VCO2 slope were significant. It must be considered that age and wedge resection were different in the subgroups of cardiovascular complications (with or without pulmonary).

Wedge resection was the most important factor (OR 0.28). Effect size for VE/VCO2 slope (OR 1.06) and age (1.04) were of much smaller magnitude.

Hence, the type of surgery (Wedge resection) could have reduced the effect size for peak VO2 and maybe that’s why it was not significant in multivariate stepwise (In univariate, the OR was 0.93). 

Also, there was a wide definition for cardiovascular complication, and this could have affected the predictive model ability to detect relevant impact of CPET variables.

Suggestions

- Review the logistic regression model, as there may be non-reported clinical characteristics that were not analysed (hypertension, diabetes, medication use).

We would like to thank the reviewer for this comment. Please note comorbidities were already part of the logistic regression as the S-MPM. Same response as to the comment No. 5 above. 

In the presented study, cardiovascular risk factors were part of the Surgical Mortality Probability Model (S-MPM), which is basically an ASA physical status adjusted for surgery risk and emergency status. Using this parameter allowed us to control the logistic regression for such a complex confounder as are comorbidities (and not creating an over-fitting model). For the reviewer, we have included frequency of the most common comorbidities in Table 2 (and also in the Supplement Table 1). As you can see, the frequency of comorbidities practically mirrors the S-MPM. We have also added a detailed explanation of components of S-MPM in the methods section.

“Surgical Mortality Probability Model (S-MPM), an ASA (American Society of Anesthesiologists) physical status adjusted for surgery risk class and emergency status (15), was used to allow patient’s co-morbidities comparison.” Line 121-124

- The major influence for cardiovascular complication risk was the type of surgery (Wedge resection). A reanalysis of logistic regression model removing these patients, may be interesting to increase the sensibility of the CPET predictive variables in the other types of surgeries.

We would like to thank the reviewer for this very interesting comment/suggestion. As suggested, we have removed patients who underwent wedge resection (n=144) and redone the multivariate logistic regression. Removing wedge resection patients decreased the number of patients with cardiovascular complications by 16. Therefore, two variables had to be removed from the logistic regression model (to prevent over-fitting model). We have removed those variables with the weakest association (FEV1/FVC and DLCO). Included variables remained age, lobectomy, thoracotomy, S-MPM, peak VO2 and VE/VCO2 slope. Out of which only VE/VCO2 slope OR=1.06 (95% CI 1.01-1.12); p=0.02 and S-MPM OR=1.46 (95% CI 1.06-1.99); p=0.02 remained significantly associated with the development of cardiovascular complications. Suggesting comorbidities and VE/VCO2 slope are the only independent predictors of cardiovascular complications in this subgroup. Wedge resections are frequent and it is possible this type of procedure will become even more frequent (Hamaji M, Miyahara S, Lee HS, Burt BM. Standardizing the time-honored wedge resection. J Thorac Dis. 2018;10(Suppl 18):S2206–8). Therefore, we have decided not to exclude these patients from our analyses in the manuscript. We hope the reviewer will agree. 

- Cardiovascular complication definition was too wide. Maybe a subgroup analysis with subdivision in minor versus major cardiovascular complications maybe more clinically relevant.

We agree with the reviewer, that dividing complications into minor and major would be interesting. However, the number of major complications (including pulmonary edema, pulmonary embolism, heart failure, acute myocardial infarction, cardiopulmonary resuscitation and stroke) was low 13 (16%) and presented “only” in 10 patients (13%). Comparison of such a small group would be hard / impossible to interpret. We hope the reviewer will agree. We comment on this in the manuscript.

“The incidence of post-operative cardiovascular complications was 22%, similar to previous reports (17,18). The most frequent cardiovascular complication was supraventricular arrhythmia (60%) in agreement with previous studies of post-lung resection surgery (18,19). The second most common complication was post-operative hypotension (58%), which is also frequent (20). Although these complications may be considered mild, post-operative arrhythmias are associated with longer hospital LOS, intensive care unit admissions, resource utilization and costs of care (17,21) and post-operative hypotension may be associated with myocardial injury (20). Indeed, in our cohort, ICU readmissions were greater and hospital and ICU LOS were significantly longer in both cardiovascular complication groups. Severe complications (except of hypotension and arrhythmia) were relatively rare (16%) and developed in only 10 patients.“ Line 230-240

Thank you for your consideration. Please direct correspondence to Ivan Cundrle Jr., Department of Anesthesiology and Intensive Care, St. Anne's University Hospital in Brno

Pekarska 53, 656 91 Brno. Phone: +420 543 182 553. E-mail address: Ivan.Cundrle@seznam.cz

Sincerely,

Ivan Cundrle Jr., M.D., Ph.D.

Department of Anesthesiology and Intensive Care

St. Anne's University Hospital in Brno

Pekarska 53, 656 91 Brno

Czech Republic

---

## [Editor Report · Decision Letter 1]

1 Aug 2022

Ventilatory efficiency is superior to peak oxygen uptake for prediction of lung resection cardiovascular complications

PONE-D-22-05505R1

Dear Dr. Cundrle Jr.,

We’re pleased to inform you that your manuscript has been judged scientifically suitable for publication and will be formally accepted for publication once it meets all outstanding technical requirements.

Kind regards,

Juliana Goulart Prata Oliveira Milani, MSc., PT

Guest Editor

PLOS ONE

Additional Editor Comments (optional):

Dear Dr. Cundrle Jr.,

First, in order to preserve transparency and uphold the integrity of the scientific process, I would like to inform the authors that, prior to my current role as a Guest Editor, I was one of the reviewers for the initial evaluation of this manuscript.

Therefore, I would like to congratulate the authors for the responses to the reviewers, for all the adjustments that were made to the manuscript, and for explanations when the suggestions could not be fully met either.

Finally, after a careful review, I am pleased to recommend the acceptance for publication. The research is well done, has an important clinical approach, and deserves publication in PLOS ONE.

Yours sincerely,

Juliana Goulart Prata Oliveira Milani

Guest Editor

---

## [Editor Report · Acceptance letter]

4 Aug 2022

PONE-D-22-05505R1 

Ventilatory efficiency is superior to peak oxygen uptake for prediction of lung resection cardiovascular complications 

Dear Dr. Cundrle Jr.:

I'm pleased to inform you that your manuscript has been deemed suitable for publication in PLOS ONE. Congratulations! Your manuscript is now with our production department. 

Kind regards, 

on behalf of

Dr. Juliana Goulart Prata Oliveira Milani 

Guest Editor

PLOS ONE